# Mouse models of hereditary hemochromatosis do not develop early liver fibrosis in response to a high fat diet

John Wagner[1], Carine Fillebeen[1], Tina Haliotis[1], Edouard Charlebois[1], Angeliki Katsarou[1], Jeannie Mui[2], Hojatollah Vali[2], Kostas Pantopoulos[1]*

**1** Lady Davis Institute for Medical Research, Jewish General Hospital, and Department of Medicine, McGill University, Montreal, Quebec, Canada, **2** Department of Anatomy and Cell Biology, McGill University, Montreal, Quebec, Canada

* kostas.pantopoulos@mcgill.ca

**Data Availability Statement:** All relevant data are within the paper.

**Funding:** This work was supported by a grant from the Canadian Institutes for Health Research (CIHR; PJT-159730).

## Abstract

Hepatic iron overload, a hallmark of hereditary hemochromatosis, triggers progressive liver disease. There is also increasing evidence for a pathogenic role of iron in non-alcoholic fatty liver disease (NAFLD), which may progress to non-alcoholic steatohepatitis (NASH), fibrosis, cirrhosis and hepatocellular cancer. Mouse models of hereditary hemochromatosis and NAFLD can be used to explore potential interactions between iron and lipid metabolic pathways. Hfe-/- mice, a model of moderate iron overload, were reported to develop early liver fibrosis in response to a high fat diet. However, this was not the case with Hjv-/- mice, a model of severe iron overload. These data raised the possibility that the *Hfe* gene may protect against liver injury independently of its iron regulatory function. Herein, we addressed this hypothesis in a comparative study utilizing wild type, Hfe-/-, Hjv-/- and double Hfe-/-Hjv-/- mice. The animals, all in C57BL/6J background, were fed with high fat diets for 14 weeks and developed hepatic steatosis, associated with iron overload. Hfe co-ablation did not sensitize steatotic Hjv-deficient mice to liver injury. Moreover, we did not observe any signs of liver inflammation or fibrosis even in single steatotic Hfe-/- mice. Ultrastructural studies revealed a reduced lipid and glycogen content in Hjv-/- hepatocytes, indicative of a metabolic defect. Interestingly, glycogen levels were restored in double Hfe-/-Hjv-/- mice, which is consistent with a metabolic function of Hfe. We conclude that hepatocellular iron excess does not aggravate diet-induced steatosis to steatohepatitis or early liver fibrosis in mouse models of hereditary hemochromatosis, irrespective of the presence or lack of Hfe.

## Introduction

Non-alcoholic fatty liver disease (NAFLD) represents the hepatic component of the metabolic syndrome (type 2 diabetes, obesity, hyperlipidemia, hypertension), and constitutes the most frequent liver disease in Western countries [1, 2]. NAFLD is characterized by excessive fat accumulation in hepatocytes in the absence of other causes of liver disease, such as alcohol

**Competing interests:** The authors have declared that no competing interests exist.

abuse or viral hepatitis. In approximately 30% of patients, NAFLD progresses from simple steatosis to non-alcoholic steatohepatitis (NASH), a chronic inflammatory condition that may further lead to liver fibrosis, cirrhosis and hepatocellular carcinoma (HCC). NAFLD patients often exhibit perturbed iron metabolism and accumulate liver iron deposits, which increases the risk for liver fibrosis [3, 4]. Consequently, manipulation of iron metabolic pathways may offer a promising therapeutic target for NAFLD [5].

Systemic iron balance is controlled by hepcidin, a liver-derived iron regulatory hormone [6]. Hepcidin limits iron efflux to the bloodstream by binding to the iron exporter ferroportin in tissue macrophages, intestinal enterocytes and other target cells, which leads to ferroportin internalization and degradation. The expression of hepcidin is induced in response to iron stores, inflammatory signals and other stimuli. Iron regulation of hepcidin involves bone morphogenetic proteins (BMPs) and the SMAD signaling cascade [7]. Genetic defects in the hepcidin pathway underlie the development of hereditary hemochromatosis, an endocrine disorder of systemic iron overload that is caused by loss of feedback regulation in iron absorption and systemic iron traffic [8, 9]. The most common form of hemochromatosis is associated with mutations in the HFE, an atypical major histocompatibility class I molecule. Inactivation of the HJV (hemojuvelin), a BMP co-receptor, leads to early onset juvenile hemochromatosis, characterized by more severe iron overload. Both HFE and HJV operate as upstream regulators of iron signaling to hepcidin [10].

Hemochromatosis patients exhibit excessive iron accumulation within hepatocytes, which predisposes them to liver fibrosis and progression to end stage liver disease [11]. Hfe-/- and Hjv-/- mice recapitulate the relatively milder or severe iron overload of patients with adult or juvenile hemochromatosis, respectively, but do not develop spontaneous early liver disease. Interestingly, Hfe-/- mice manifested a NASH-like phenotype and early liver fibrosis after feeding a high fat diet for 8 weeks, which was only partially attributed to iron [12]. On the other hand, a 12-week high fat diet intake caused liver steatosis but did not promote steatohepatitis or liver fibrosis in Hjv-/- mice, in spite of severe hepatocellular iron overload [13]. These data raised the possibility for a protective function of Hfe against metabolic liver disease. Herein, we explored this hypothesis by comparing pathophysiological responses of mice with single or combined Hfe/Hjv deficiency to high fat diet.

## Materials and methods

### Animals

Hfe-/- and Hjv-/- mice were provided by Dr. Nancy Andrews (Duke University) and were backcrossed for 10 generations to the C57BL/6J genetic background. Double Hfe-/-Hjv-/- mice were obtained by intercrossing single Hfe-/- and Hjv-/- animals [14]; the breeding also yielded wild type C57BL/6J mice, which were used as control. Four-week old male animals (n = 10 for each genotype) were fed after weaning for 14 weeks with a standard diet containing 200 ppm iron (Harlad Teklad 2018), or high fat diets containing either 50 or 200 ppm iron (Harlad Teklad TD.88137 or TD.180571, respectively). The exact composition of the Harlad Teklad 2018 and TD.88137 diets can be found in https://www.envigo.com/resources/data-sheets/2018-datasheet-0915.pdf and http://www.envigo.com/resources/data-sheets/88137.pdf, respectively. The TD.180571 diet was identical to TD.88137, except that it was supplemented with extra iron to match the 200 ppm iron content of the standard Harlad Teklad 2018 diet. A group of 10-week old male wild type mice were treated for 6 weeks with $CCl_4$ to develop liver fibrosis [15]. All animals were on C57BL/6J genetic background. They were housed in a temperature-controlled environment (22 ± 1˚ C, 60 ± 5% humidity), with a 12-hour light/dark cycle and were allowed *ad libitum* access to diets and drinking water. At the endpoint, the

mice were sacrificed by $CO_2$ inhalation, followed by cardiac puncture to collect blood, and cervical dislocation. Blood was obtained by cardiac puncture and clotted at room temperature for 1 h. Serum was separated by centrifugation (2000 g for 10 min), snap frozen in liquid nitrogen and stored at -80˚C for biochemical analysis. Livers were rapidly excised and tissue sections were snap frozen in liquid nitrogen and stored at -80˚C for biochemical studies. Other liver sections were fixed in 10% neutral-buffered formalin and embedded in paraffin for histological and ultrastructural studies. All animal procedures were approved by the Animal Care Committee of McGill University (protocol 4966).

### Serum biochemistry

Iron, transferrin saturation, total iron binding capacity (TIBC), glucose, triglycerides, cholesterol, HDL cholesterol, alanine transaminase (ALT) and aspartate transaminase (AST) were measured with a Roche Hitachi 917 Chemistry Analyzer at the Biochemistry Department of the Jewish General Hospital.

### Liver biochemistry

Liver extracts were analyzed for non-heme iron by the ferrozine assay [16], and for the presence of collagen by a colorimetric hydroxyproline assay (QuickZyme Biosciences), according to the manufacturer's recommendations.

### Histopathology and immunohistochemistry

Deparaffinized liver sections were stained with hematoxylin and eosin (H&E), Perls' Prussian blue or Masson's trichrome to assess tissue architecture, iron deposits or collagen, respectively. Expression of α-smooth muscle actin (α-SMA) was analyzed by immunohistochemistry, as previously described [15].

### Transmission electron microscopy

Liver sections were prepared for ultrastructural studies and analyzed by transmission electron microscopy (TEM) as described [13].

### Statistics

Statistical analysis was performed with the GraphPad Prism software (v. 5.0e). Quantitative data are expressed as mean ± standard error of the mean (SEM). Statistical analysis across multiple groups (genotypes and diets) was performed by two-way ANOVA with Bonferroni post-test comparison. A probability value $p < 0.05$ was considered statistically significant.

## Results

### Responses of mouse models of hemochromatosis to high fat diets

Hfe-/- mice represent a model of the most common and relatively milder form of hemochromatosis, while Hjv-/- and double Hfe-/-Hjv-/- mice develop more severe iron overload [14]. Thus, when compared to isogenic wild type control animals, serum iron and transferrin saturation were modestly elevated in Hfe-/- and substantially augmented in Hjv-/- and double Hfe-/-Hjv-/- mice on a standard diet (Fig 1A and 1B). TIBC appeared slightly reduced in all hemochromatosis models (Fig 1C). Feeding a high fat diet (HFD) containing 50 ppm iron immediately after weaning for 14 weeks resulted an approximately 35% body weight gain in all mice irrespective of genotype, as compared to age- and sex-matched controls fed a standard

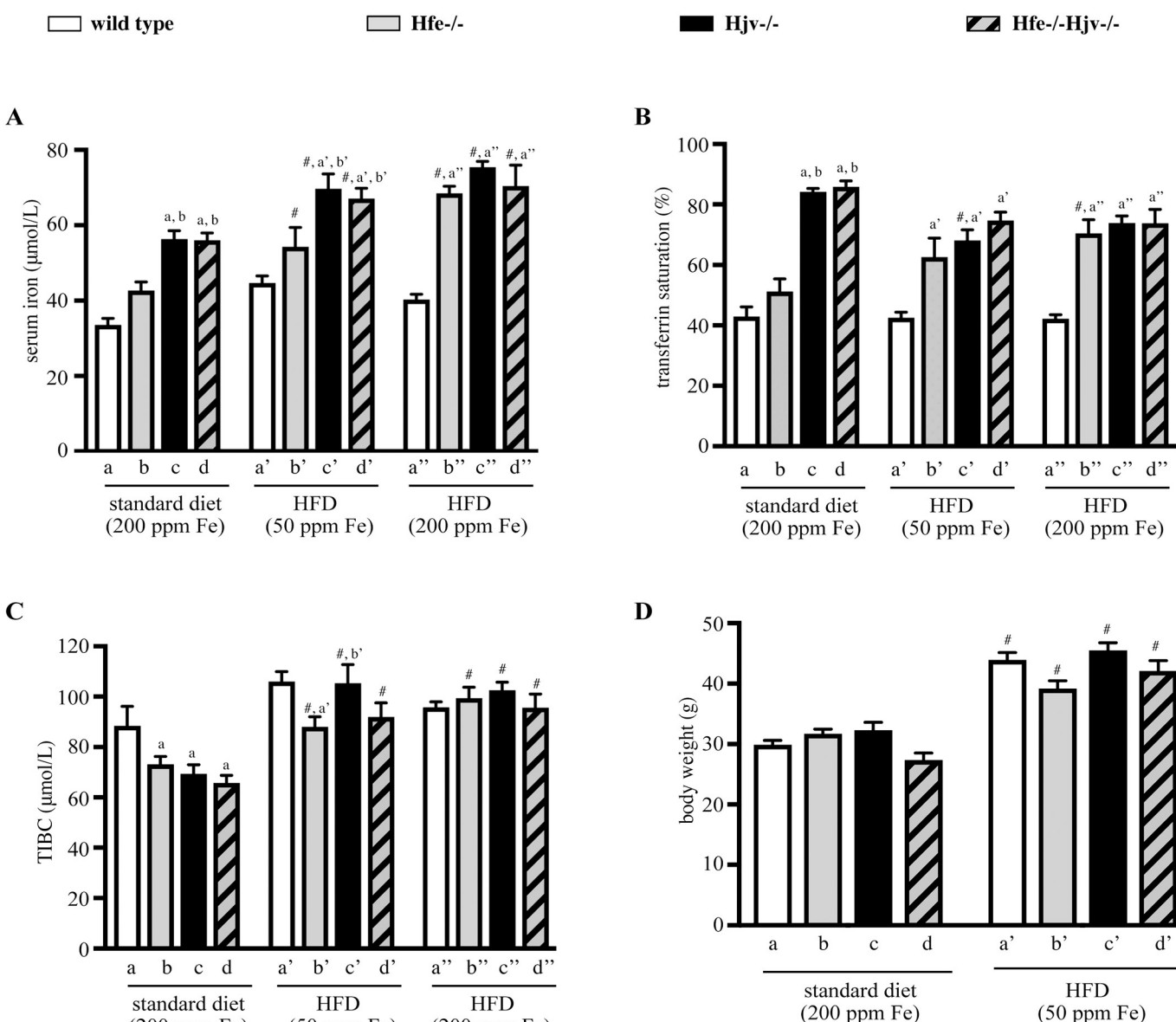

**Fig 1. Effects of high fat diets (HFDs) on serum iron parameters and body weight in mouse models of hemochromatosis.** Wild type control (wt), Hfe-/-, Hjv-/- and double Hfe-/-Hjv-/- mice (n = 10 per group, all males in C57BL/6 genetic background) were placed immediately after weaning on a standard diet containing 200 ppm iron, or on HFDs containing either 50 or 200 ppm iron. After 14 weeks, body weights were monitored, the animals were sacrificed, and sera and liver tissues were obtained for analysis. (A) serum iron; (B) transferrin saturation; (C) total iron binding capacity (TIBC); and (D) body weights of the mice. All data are presented as the mean ± SEM. Statistical analysis was performed by two-way ANOVA. Statistically significant differences ($p<0.05$) across genotypes (versus columns a, b, a', b', c', a") are indicated by a, b, a', b', c', a". Statistically significant differences ($p<0.05$) within each genotype compared to standard diet are indicated by #.

diet (Fig 1D). A similar weight gain was also noted in all mice fed the HFD supplemented with extra iron to match the 200 ppm iron content of the standard diet; nevertheless, the actual weights were not registered. HFD intake (with either 50 or 200 ppm iron) was associated with slight increases in serum iron (Fig 1A) and TIBC (Fig 1C), and commensurate drops in transferrin saturation (Fig 1B).

In addition, HFD intake promoted an increase in levels of serum glucose, cholesterol and HDL-cholesterol, but not triglycerides, in all genotypes (Fig 2A–2D). The effects on cholesterol

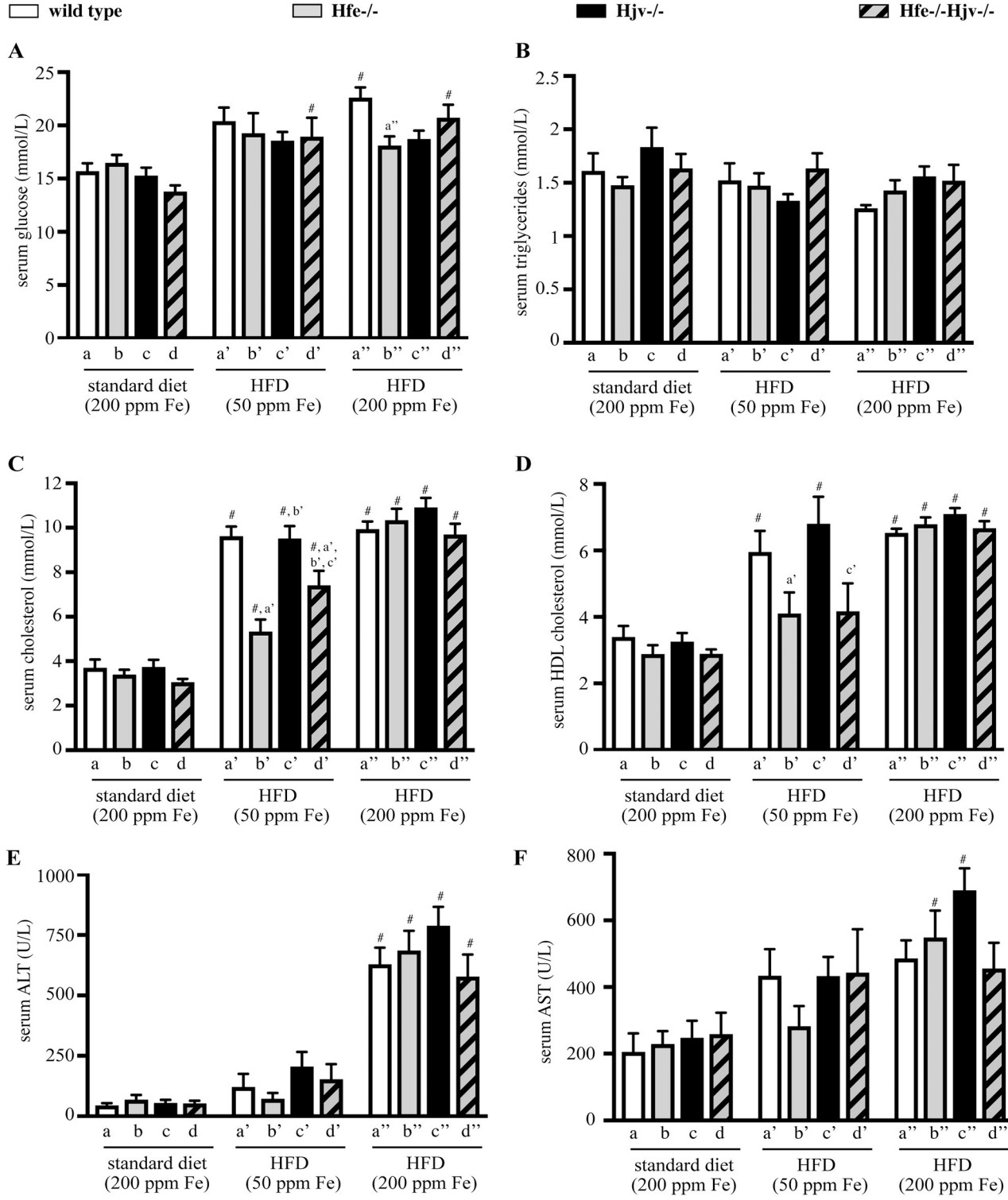

**Fig 2. Effects of high fat diet on serum biochemistry.** Sera from mice described in Fig 1 were analysed for: (A) glucose; (B) triglycerides; (C) cholesterol; (D) HDL-cholesterol; (E) ALT; and (F) AST. All data are presented as the mean ± SEM. Statistical analysis was performed by two-way ANOVA. Statistically significant differences (p<0.05) across genotypes (versus columns a', b', c', a") are indicated by a', b', c', a". Statistically significant differences (p<0.05) within each genotype compared to standard diet are indicated by #.

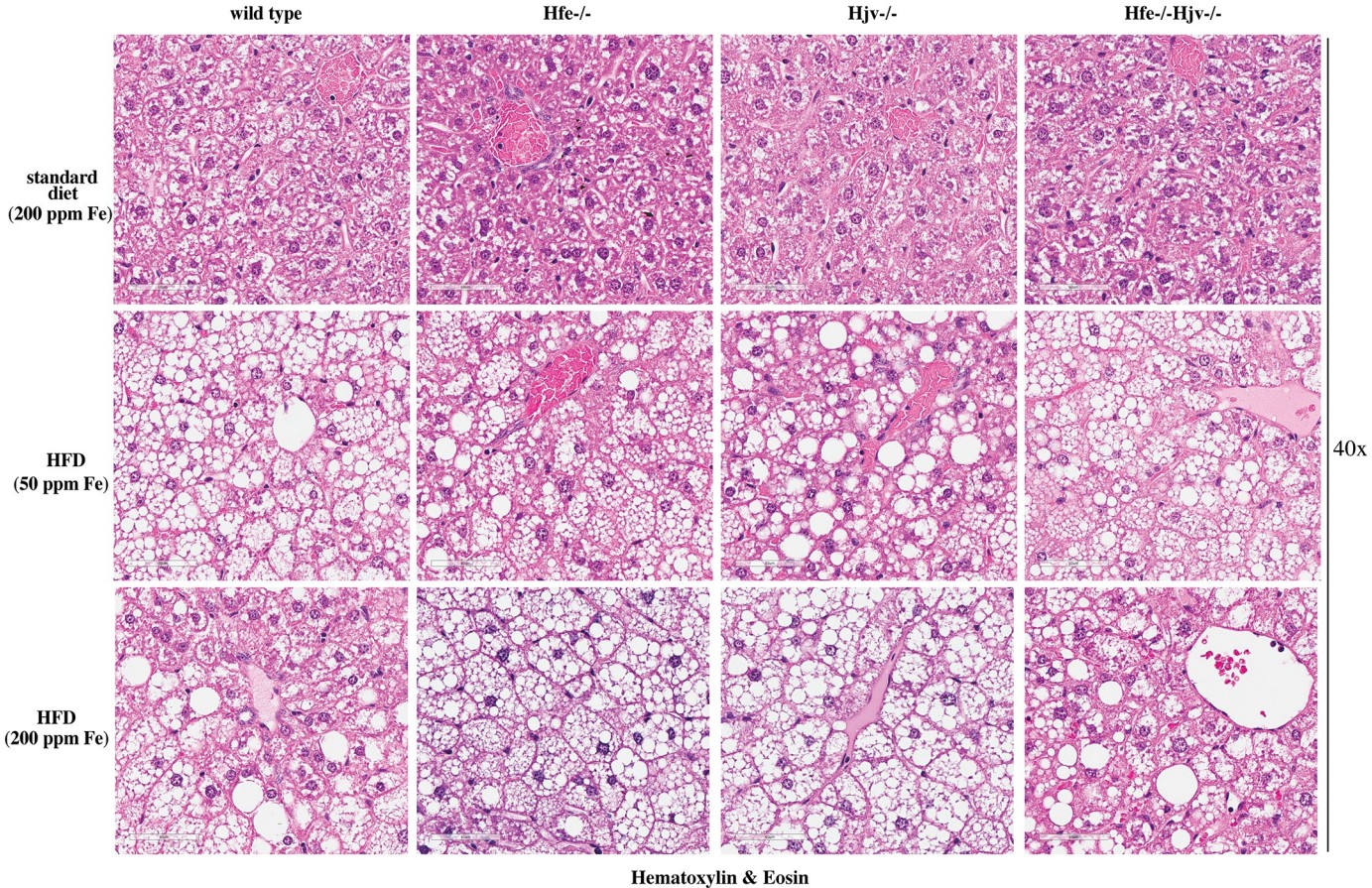

**Fig 3. High fat diet promotes liver steatosis without necroinflammation in mouse models of hemochromatosis.** Liver sections from the mice described in Fig 1 were stained with H&E. Original magnification: 40x.

and HDL-cholesterol appeared less pronounced in Hfe-/- mice on HFD with 50 ppm iron. We also analyzed serum transaminases ALT and AST. Thus, the HFD with 200 ppm iron triggered a profound (>10-fold) increase in ALT, and a relatively more modest (~50–75%) increase in AST levels in all genotypes (Fig 2E and 2F).

Livers of all mice on HFD (with either 50 or 200 ppm iron) were enlarged and exhibited mixed micro- and macrovesicular steatosis (Fig 3). There were no differences among genotypes in the degree and the pattern of fat deposition. Histological analysis did not reveal any large inflammatory foci or clusters of immune cells in any of the examined livers. The above data suggest that the HFDs promote similar obesity and liver steatosis in wild type mice and mouse models of hemochromatosis, irrespectively of their iron content.

### Does hepatic iron overload trigger progression of steatosis to liver fibrosis?

As expected, mouse models of hemochromatosis developed hepatic iron overload (Fig 4). This was relatively mild in Hfe-/- mice and more intense in Hjv-/- and double Hfe-/-Hjv-/- counterparts, in agreement with earlier data [14]. Liver iron accumulation was generally reduced in animals fed the HFDs, as previously observed [12, 13, 17, 18], but Hfe-/- mice did not follow this trend. The decreases in liver iron content were more pronounced in Hjv-/- and double Hfe-/-Hjv-/- mice on the HFD containing 50 ppm iron (Fig 4A). Staining of liver sections with

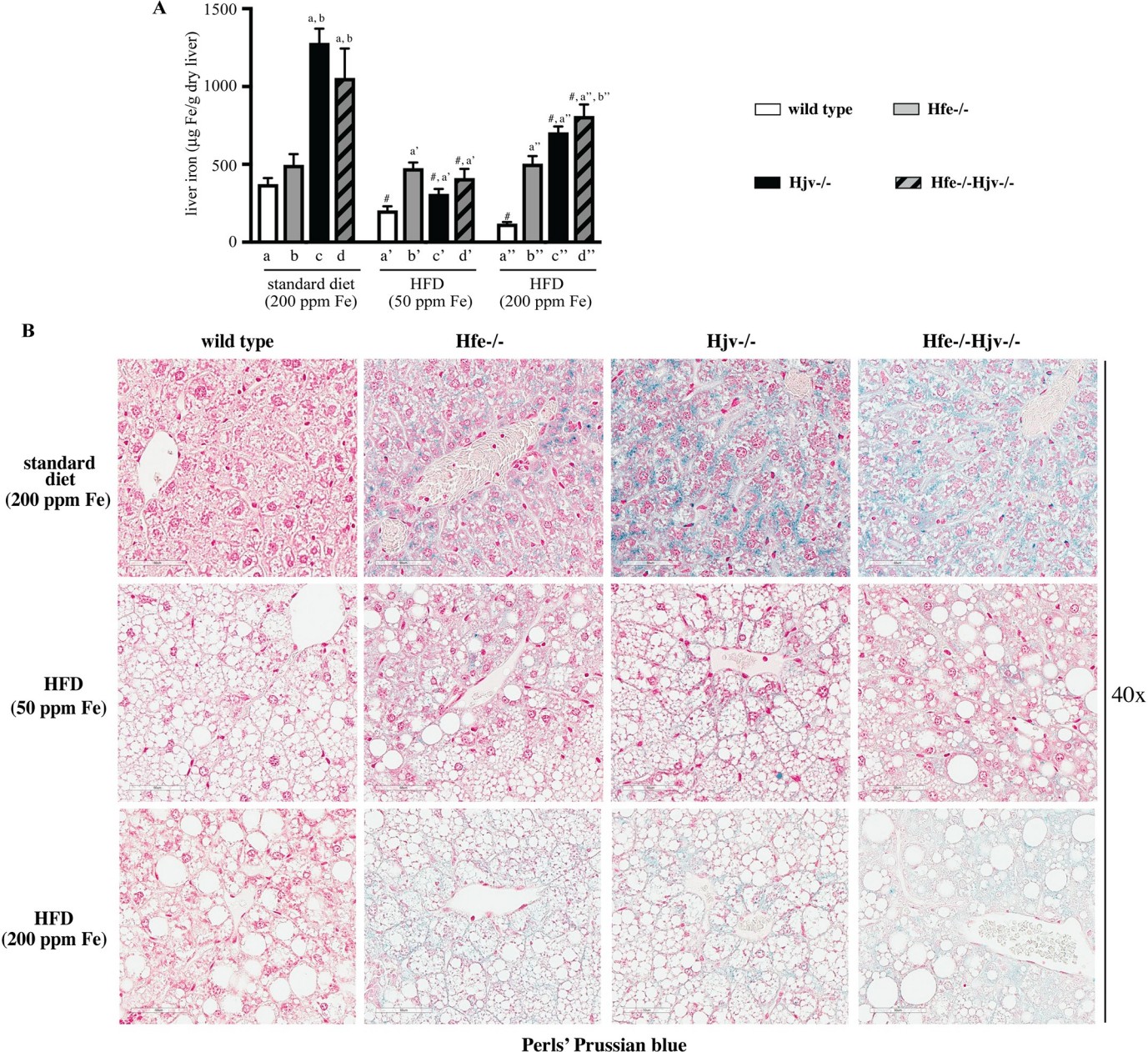

**Fig 4. Evaluation of hepatic iron content in mouse models of hemochromatosis on standard or high fat diets.** Livers from the mice described in Fig 1 were used to quantify and histologically assess iron content. (A) Quantification of non-heme iron by the ferrozine assay. Data are presented as the mean ± SEM. Statistical analysis was performed by two-way ANOVA. Statistically significant differences ($p < 0.05$) across genotypes (versus columns a, b, a', a", b") are indicated by a, b, a', a", b". Statistically significant differences ($p < 0.05$) within each genotype compared to standard diet are indicated by #. (B) Histological detection of iron deposits by staining with Perls' Prussian blue. Original magnification: 40x.

Perls' Prussian blue revealed the presence of iron deposits in hepatocytes of hemochromatotic but not wild type control mice (Fig 4B). The signal was diluted in liver sections from animals fed the HFDs, and its intensity was proportional to the iron content of the HFD.

Hepatocellular iron overload was not associated with liver fibrosis in any of the mice fed standard diet or HFDs, independently of iron content. The presence of collagen was assessed histologically by Masson's trichrome staining (Fig 5A), and biochemically by the hydroxyproline

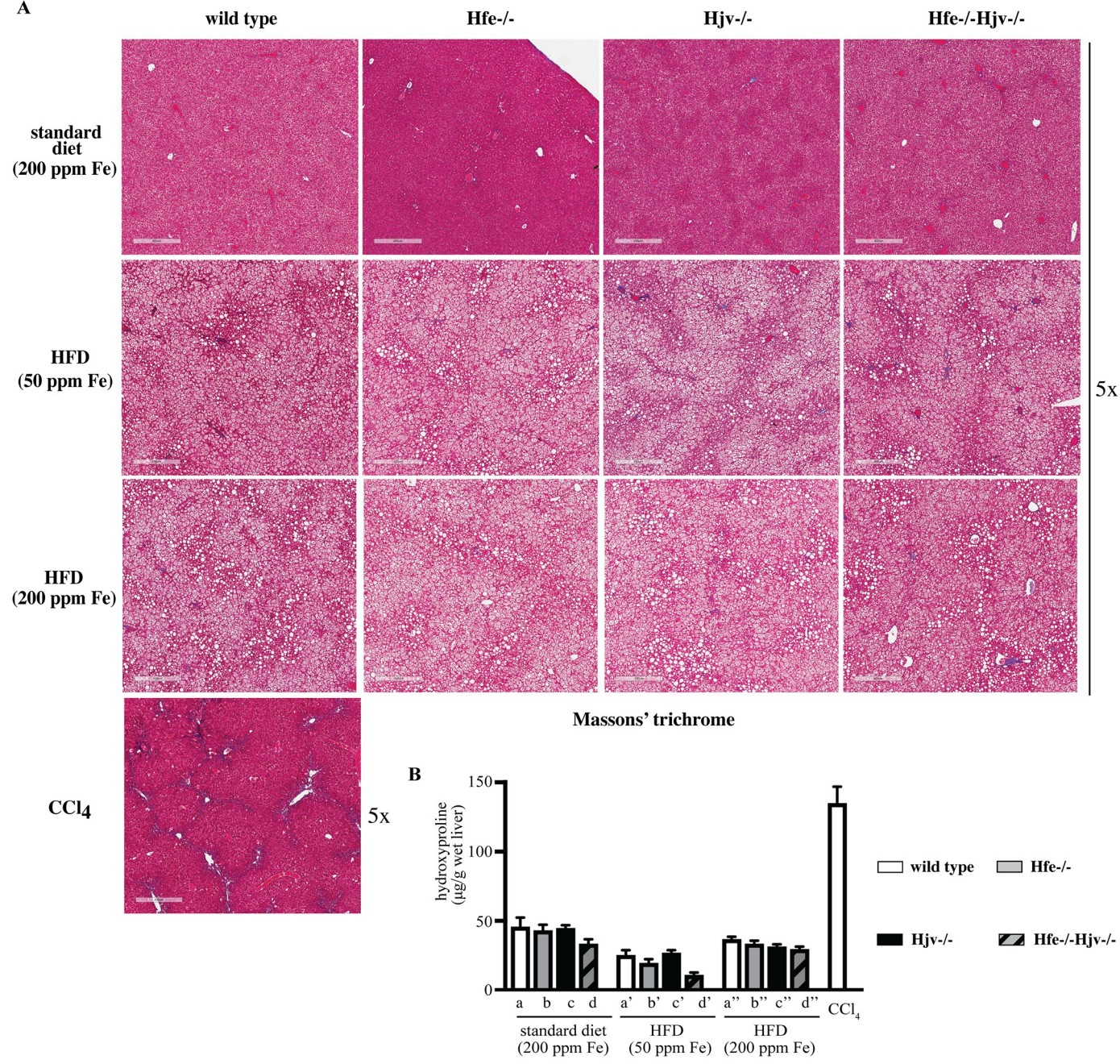

**Fig 5. High fat diet does not promote liver fibrosis in mouse models of hemochromatosis.** Livers from the mice described in Fig 1 were used to histologically assess fibrosis and to quantify collagen. Livers from mice treated with $CCl_4$ were used as positive control. (A) Staining with Masson's trichrome. Original magnification: 5x. (B) Quantification of collagen by the hydroxyproline assay. Data are presented as the mean ± SEM. Statistical analysis was performed by two-way ANOVA.

assay (Fig 5B). Liver samples from $CCl_4$-treated wild type mice served as positive control for liver fibrosis. Immunohistochemical analysis showed expression of α-SMA, a marker of activated hepatic stellate cells, in livers of Hjv-/-, Hfe-/-Hjv-/- and to a smaller extent also Hfe-/- mice (Fig 6). In addition, HFD intake (50 ppm iron) promoted α-SMA expression in livers of wild type mice. Thus, either hepatic iron overload or steatosis appear to activate fibrogenic

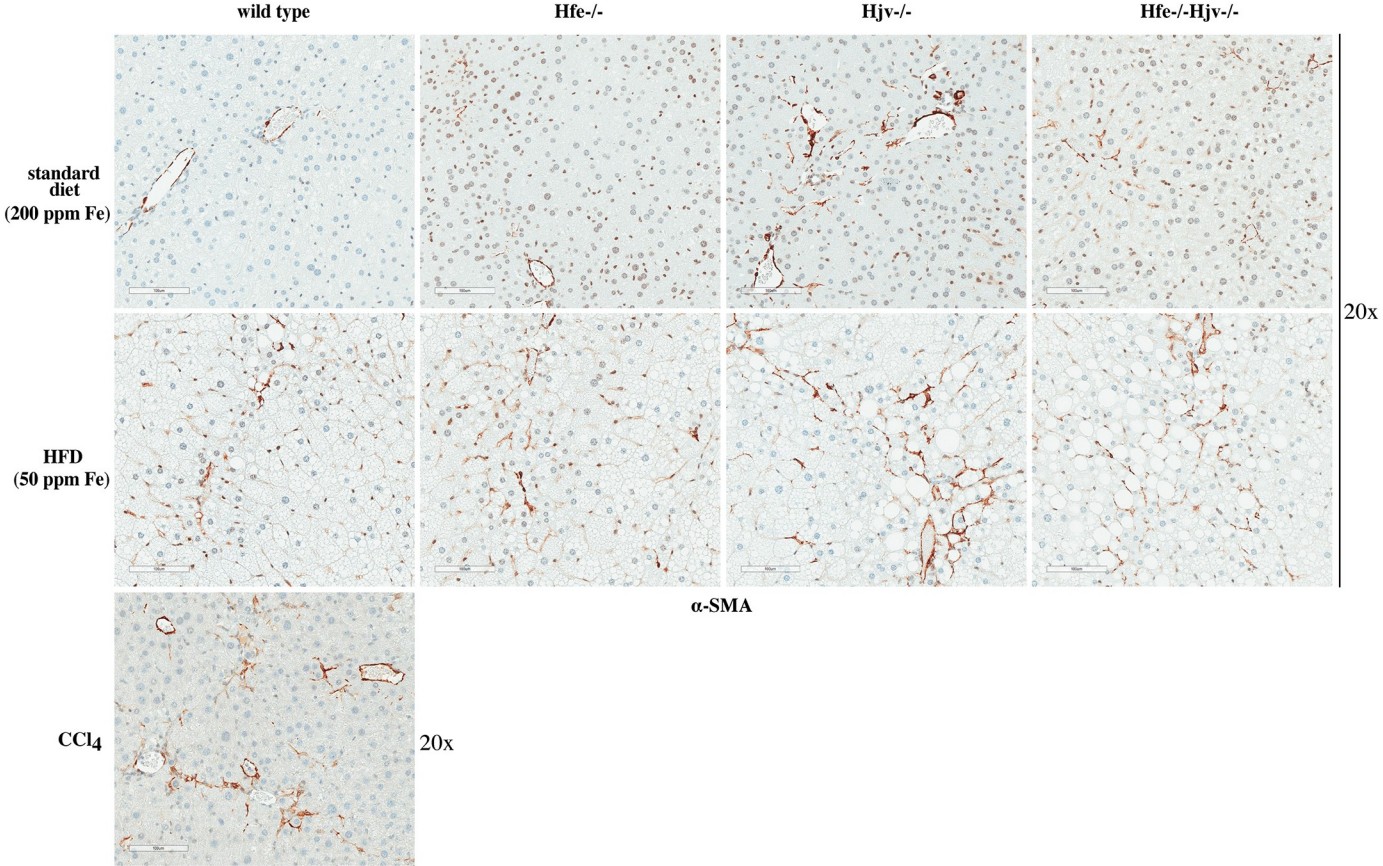

**Fig 6. Hepatic iron overload or steatosis trigger activation of hepatic stellate cells.** Liver sections from mice described in Figs 1 and 5 were used for immunohistochemical detection of α smooth muscle actin (α-SMA), a marker of hepatic stellate cell activation. Original magnification: 20x.

responses in mouse models of hemochromatosis; however, their combination does not dramatically accelerate them to promote rapid progression of liver disease.

## Ultrastructural studies

Analysis of the liver tissue by TEM corroborated the absence of inflammation or fibrosis in hemochromatotic and steatotic mice. Data obtained from animals on standard diet are shown in Fig 7 (top). Wild type hepatocytes exhibited a normal architecture and contained regular lipid droplets and glycogen granules. On the other hand, some mitochondria displayed darker contrast suggesting the possible presence of lipids stained by osmium tetroxide. The same feature was observed in Hfe-/- hepatocytes, but some mitochondria were swollen (arrow). Hjv-/- hepatocytes displayed well-preserved mitochondria with much less dense glycogen and reduced lipid droplets. ER appears to be less organized and located more around mitochondria. Interestingly, in Hfe-/-Hjv-/- hepatocytes the reduced lipid content persisted but glycogen density and mitochondrial appearance were normalized.

Data from mice on HFD (50 ppm iron) are depicted in Fig 7 (bottom). The number and size of lipid droplets was increased in all genotypes. Nevertheless, the number of fat vesicles was relatively higher in Hfe-/-, and the overall fat content lower in Hjv-/- hepatocytes. The appearance of mitochondria was improved in Hfe-/- hepatocytes but the glycogen density was reduced compared to that from mice on standard diet. By contrast, some mitochondria looked

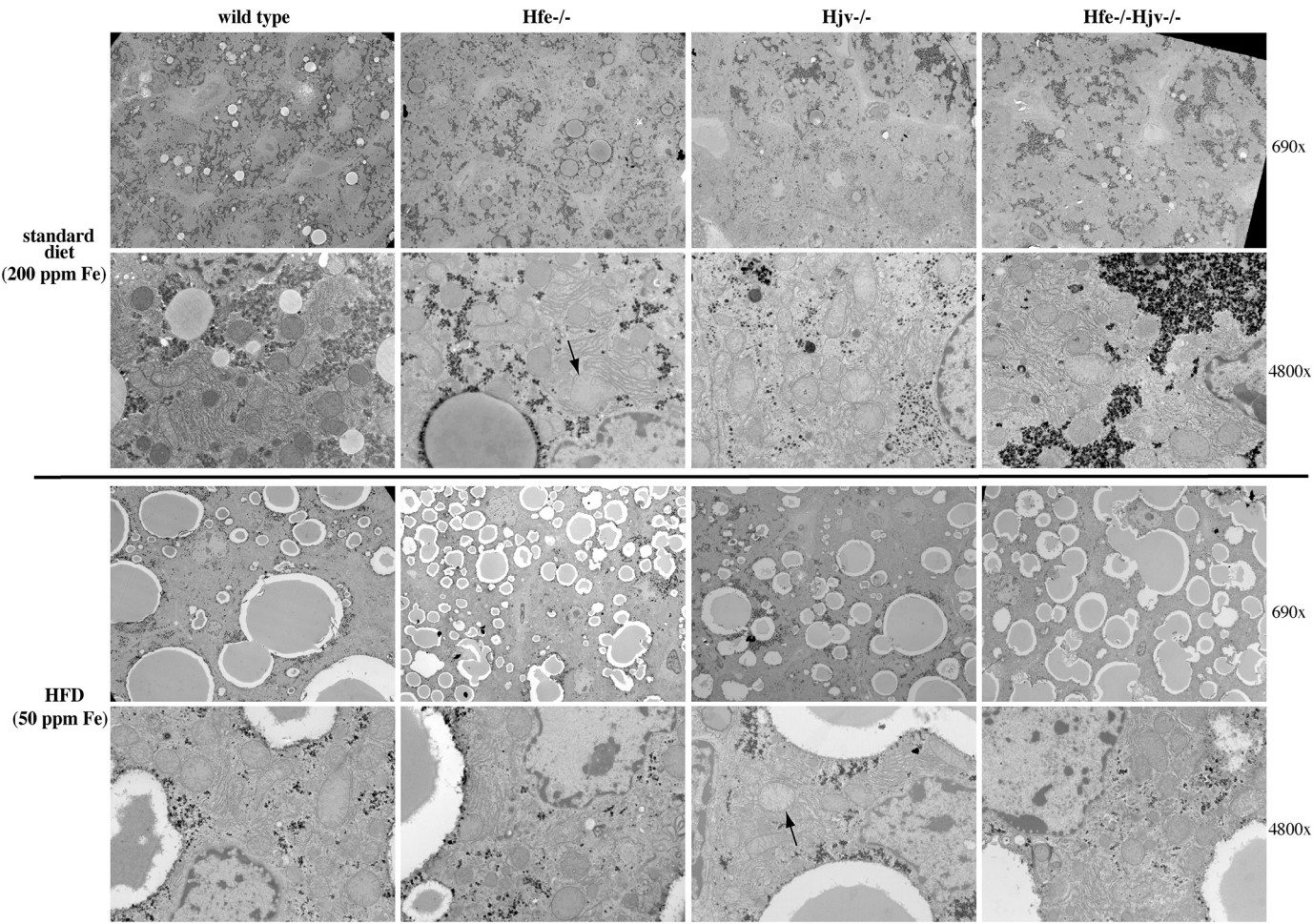

**Fig 7. Ablation of Hfe or Hjv is associated with ultrastructural alterations in hepatocytes.** Liver sections from mice described in Fig 1 were analyzed by transmission electron microscopy (TEM). Two magnifications (690x and 4800x) are shown for each sample. Arrows indicate mitochondria with abnormal morphology.

abnormal in Hjv-/- hepatocytes (arrows), the glycogen density remained low, and the ER got disorganized in response to the HFD. With respect to mitochondria and ER, hepatocytes from Hfe-/-Hjv-/- and Hfe-/- mice on HFD had a similar phenotype. Glycogen density appears reduced. Taken together, the transmission electron microscopy data reveal distinct ultrastructural features among mouse models of hemochromatosis and raise the possibility for metabolic functions of Hfe and Hjv.

## Discussion

Excessive accumulation of iron or fat in hepatocytes are known pathogenic factors for liver injury. Clinical data suggested a synergism of hepatocellular iron overload and steatosis in liver disease progression to liver fibrosis [19–21]. We sought to validate these findings using Hjv-/- mice, a model of severe, early onset juvenile hemochromatosis. However, feeding Hjv-/- mice with a HFD did not cause progression of liver steatosis to steatohepatitis or fibrosis, in spite of profound iron overload [13]. In another study, intake of a HFD triggered early liver fibrosis in Hfe-/- mice, a model of relatively mild, late onset adult hemochromatosis [12].

Hfe-/- mice also manifested exacerbated hepatotoxicity following combined HFD and alcohol intake [22].

Based on these findings, we reasoned that Hfe may exert a potential iron-independent function in liver disease progression and utilized herein double Hfe-/-Hjv-/- mice to explore this hypothesis. Hfe-/-Hjv-/- and Hjv-/- mice exhibit an indistinguishable iron overload phenotype, suggesting that the *Hjv* gene is epistatic to *Hfe* in iron homeostasis [14]. Thus, comparing the responses of these animals to a HFD would provide insights on the role of Hfe in metabolic liver disease. To this end, isogenic sex and age-matched wild type control, Hfe-/-, Hjv-/- and double Hfe-/-Hjv-/- mice were exposed for 14 weeks to HFDs only differing in iron content. HFD intake triggered obesity and liver steatosis in all genotypes but did not promote steatohepatitis or liver fibrosis in any of them (Figs 3 and 5), independently of iron content. These data do not support an iron-independent protective role of Hfe against liver disease progression.

The Hfe-/- mice used in this work and in [12] shared the same genetic background (C57BL/6J). In our experiments, exposure to the HFD started a bit earlier (immediately after weaning vs at the 6th week of age) and lasted considerably longer (14 vs 8 weeks); this setting is expected to be more favorable to liver injury. Thus, we speculate that the discordant results reported herein and in [12] may be related to a genetic drift in mouse colonies or to small variations in the HFD content. It will be interesting to examine responses of Hfe-/-Hjv-/- and Hjv-/- mice to the HFD used in [12].

The data with Hjv-/- mice are largely consistent with our previous findings [13]. However, we did not observe reduced body weight gain of Hjv-/- compared to wild type mice in response to HFD (Fig 1A), contrary to data in [13]. Presumably, this discrepancy is related to the different timing the mice were placed on HFD (immediately after weaning vs at the 10th week of age) and the duration of HFD feeding (14 vs 12 weeks).

Our data suggest that hepatocellular iron overload does not aggravate liver steatosis to steatohepatitis or fibrosis in the mouse models of hemochromatosis, at least within the time frame of 14 weeks. Nevertheless, it contributes to liver injury, as reflected in the increased serum ALT and AST values of mice fed the HFD containing 200 ppm iron (Fig 1E and 1F). It is conceivable that iron's hepatotoxicity is manifested histologically at later time points. It should be noted that either iron or fat accumulation trigger fibrogenic responses on their own right, as illustrated by the induction of α-SMA (Fig 6), a marker of hepatic stellate cell activation to collagen-secreting myofibroblasts [23]. However, their combination does not appear to aggravate or accelerate these responses in mice.

Notably, HFD intake appeared to mitigate hepatic iron overload (Fig 4), presumably due to reduced iron absorption, and in agreement with earlier data [12, 13, 17, 18]. Nevertheless, HFD-fed hemochromatotic mice of our study had pathological hepatic iron content, which was proportional to the iron content of the diet and was visible by Perls staining (Fig 4B). Considering the ~40% increased size of the steatotic livers, the quantitative ferrozine assay as expressed in μg of iron per gram of dry liver (Fig 4A) probably underestimates the true hepatocellular iron concentration. In any case, the increased hepatic iron content that was achieved in animals fed the iron-supplemented HFD did not enhance liver pathology. Along these lines, we previously reported that Hjv-/- mice were resistant to liver fibrosis even after feeding a HFD containing 2% carbonyl iron; under these conditions, the excessive iron content of the HFD resulted in reduced steatosis [13].

Assuming that even a low degree of hepatocellular iron overload is often pathogenic in humans, our results do not recapitulate clinical data on NAFLD in hemochromatosis patients [19–21]. This notion underlines known shortcomings of mouse models. Thus, contrary to hemochromatosis patients, genetic mouse models of hemochromatosis are spared from spontaneous liver injury due to hepatocellular overload, unless they are subjected to

chronic (1 year) feeding with high iron diets [24, 25]. On the other hand, dietary iron over-load was shown to promote steatohepatitis in genetically obese Lepr[db/db] mice; notably excess iron accumulated predominantly in reticuloendothelial cells [26]. This result is in line with clinical studies establishing a pathogenic role of reticuloendothelial iron overload, which has been documented in many NAFLD patients [27–29].

On a final note, the TEM analysis in Fig 7 uncovers marked differences in hepatocyte ultra-structure of Hfe-/- and Hjv-/- mice. It identified apparent mitochondrial abnormalities in Hfe-/- hepatocytes, and confirmed the previously reported [13] reduction in fat and glycogen stores in Hjv-/- hepatocytes. Strikingly, glycogen density and mitochondrial morphology were corrected in Hfe-/-Hjv-/- hepatocytes. These findings are consistent with metabolic functions of Hfe and Hjv, most likely unrelated to their iron regulatory activities. Future experiments, for instance by using high-throughput -omics technologies, could shed more light on relevant biochemical mechanisms and pathophysiological implications.

## Acknowledgments

We thank Drs. Naciba Benlimame for assistance with histology. This work was supported by a grant from the Canadian Institutes for Health Research (CIHR; PJT-159730).

## Author Contributions

**Conceptualization:** Kostas Pantopoulos.

**Investigation:** John Wagner, Carine Fillebeen, Edouard Charlebois, Angeliki Katsarou.

**Methodology:** Tina Haliotis, Jeannie Mui, Hojatollah Vali.

**Supervision:** Kostas Pantopoulos.

**Visualization:** Tina Haliotis.

**Writing – original draft:** Kostas Pantopoulos.

**Writing – review & editing:** Kostas Pantopoulos.

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
