## [Decision Letter · Decision Letter 0]

8 Aug 2019

Mouse models of hereditary hemochromatosis do not develop early liver fibrosis in response to a high fat diet

PONE-D-19-17283

Dear Dr. Pantopoulos,

We are pleased to inform you that your manuscript has been judged scientifically suitable for publication and will be formally accepted for publication once it complies with all outstanding technical requirements.

With kind regards,

Fanis Missirlis, Ph.D.

Academic Editor

PLOS ONE

JOURNAL REQUIREMENTS:

Please address the following in your final submission:

1. Please include in your methods section, the source of the mice used in this study.

2. Please also include whether you used a second confirmatory method of euthanasia in addition to CO2 inhalation.

Reviewers' comments:

Reviewer's Responses to Questions

**Comments to the Author**

1. Is the manuscript technically sound, and do the data support the conclusions?

Reviewer #1: Yes

Reviewer #2: Yes

2. Has the statistical analysis been performed appropriately and rigorously? 

Reviewer #1: Yes

Reviewer #2: Yes

3. Have the authors made all data underlying the findings in their manuscript fully available?

Reviewer #1: Yes

Reviewer #2: Yes

4. Is the manuscript presented in an intelligible fashion and written in standard English?

Reviewer #1: Yes

Reviewer #2: Yes

5. Review Comments to the Author

Reviewer #1: This is the revised manusctipt by changing the experimental condition, notaby that iron content in high fat diet was standerized. Accordingly, the results in three groups were comparable. The discussion was carefully presented to avoid overspeculation.

Reviewer #2: Summary: This study addresses the effect of a high fat diet given to three different mouse models of hereditary hemochromatosis on liver health. The findings, that iron overload does not straight forward correlate with liver damage in dietary induced liver steatosis are unexpected and intriguing. Further, the findings on modulated fat and glycogen content of hepatocytes of Hfe or Hjv KO mice, and a possible connection to mitochondrial morphology will lead to further interesting research.

Figures 1 and 2 are in very low quality and the letters indicating statistical significance are hard to see. Also, in the figure legends, where it says:” Statistically significant differences (p<0.05) across genotypes (versus columns a, b, a’, b’, c’, a’’) are indicated by a, b, a’, b’, c’, a’’ and across diets by #” it is not clear, what the # exactly stands for. For example, the # on column c’ in figure 1A. Does it mean that this column is statistically different from column c or c” or both?

Actually, all figures are in such low quality, that I cannot assess them well. From a superficial view, it seems that the conclusions are supported by the presented data. Possibly, some quantitative image analysis, especially on the activated stellate cells could give additional strength to the presented interpretations.

6. PLOS authors have the option to publish the peer review history of their article (what does this mean?). If published, this will include your full peer review and any attached files.

Reviewer #1: No

Reviewer #2: No

---

## [Editor Report · Acceptance letter]

14 Aug 2019

PONE-D-19-17283 

Mouse models of hereditary hemochromatosis do not develop early liver fibrosis in response to a high fat diet 

Dear Dr. Pantopoulos:

I am pleased to inform you that your manuscript has been deemed suitable for publication in PLOS ONE. Congratulations! Your manuscript is now with our production department. 

With kind regards,

on behalf of

Dr. Fanis Missirlis 

Academic Editor

PLOS ONE